# Application of the Dynamical Network Biomarker Theory to Raman Spectra

**DOI:** 10.3390/biom12121730

**Published:** 2022-11-22

**Authors:** Takayuki Haruki, Shota Yonezawa, Keiichi Koizumi, Yasuhiko Yoshida, Tomonobu M. Watanabe, Hideaki Fujita, Yusuke Oshima, Makito Oku, Akinori Taketani, Moe Yamazaki, Taro Ichimura, Makoto Kadowaki, Isao Kitajima, Shigeru Saito

**Affiliations:** 1Faculty of Sustainable Design, University of Toyama, Toyama 930-8555, Japan; 2Research Center for Pre-Disease Science, University of Toyama, Toyama 930-8555, Japan; 3Division of Presymptomatic Disease, Institute of Natural Medicine, University of Toyama, Toyama 930-0194, Japan; 4Graduate School of Science and Engineering, University of Toyama, Toyama 930-8555, Japan; 5RIKEN Center for Biosystems Dynamics Research, Kobe 650-0047, Japan; 6Research Institute for Radiation Biology and Medicine, Hiroshima University, Hiroshima 734-8553, Japan; 7Faculty of Engineering, University of Toyama, Toyama 930-8555, Japan; 8Institute for Open and Transdisciplinary Research Initiatives, Osaka University, Suita 565-0871, Japan; 9Faculty of Medicine, University of Toyama, Toyama 930-0194, Japan

**Keywords:** dynamical network biomarker (DNB) theory, Raman spectra, Raman spectroscopy, T cell activation, transition state

## Abstract

The dynamical network biomarker (DNB) theory detects the early warning signals of state transitions utilizing fluctuations in and correlations between variables in complex systems. Although the DNB theory has been applied to gene expression in several diseases, destructive testing by microarrays is a critical issue. Therefore, other biological information obtained by non-destructive testing is desirable; one such piece of information is Raman spectra measured by Raman spectroscopy. Raman spectroscopy is a powerful tool in life sciences and many other fields that enable the label-free non-invasive imaging of live cells and tissues along with detailed molecular fingerprints. Naïve and activated T cells have recently been successfully distinguished from each other using Raman spectroscopy without labeling. In the present study, we applied the DNB theory to Raman spectra of T cell activation as a model case. The dataset consisted of Raman spectra of the T cell activation process observed at 0 (naïve T cells), 2, 6, 12, 24 and 48 h (fully activated T cells). In the DNB analysis, the F-test and hierarchical clustering were used to detect the transition state and identify DNB Raman shifts. We successfully detected the transition state at 6 h and related DNB Raman shifts during the T cell activation process. The present results suggest novel applications of the DNB theory to Raman spectra ranging from fundamental research on cellular mechanisms to clinical examinations.

## 1. Introduction

The dynamical network biomarker (DNB) theory [1,2] was developed to detect early warning signals [3,4,5,6,7] at the transition state just before a state transition, such as the pre-disease state before the transition from a healthy state to a disease state. The DNB theory has been applied to gene expression in several diseases, such as lung injury [1,8,9,10], liver cancer [1,9], breast cancer [9,10], influenza infection [10], type 1 and 2 diabetes [11,12], metabolic syndrome [13] and hepatocellular carcinoma [8,14]. A landscape DNB (l-DNB) with a single-sample network has recently been proposed [10,15,16,17]. The key concept of the DNB theory is that the stability of the target system gradually decreases before the transition and a subset of strongly correlated system variables begins to fluctuate due to increased susceptibility to disturbances. These abnormal fluctuations peak at the transition state, namely, the state just before a transition. Therefore, by focusing on fluctuations and correlations, particularly at the transition state, it is possible to detect the early warning signals of an upcoming transition. It is important to note that the DNB theory is not limited to the detection of pre-disease states; it may also be applied in principle to many other cases in which the target system exhibits any type of transition and multivariate time-series data throughout the transition process may be obtained [18].

Although the DNB theory has been applied to gene expression in several diseases, destructive testing by microarrays is a critical issue. Therefore, other biological information obtained by non-destructive testing is desirable; one such piece of information is Raman spectra measured by Raman spectroscopy.

Raman spectroscopy, a vibrational spectroscopic technique that is widely used to analyze molecular compositions in biological specimens (e.g., living cells and tissues), has great potential for the detection of intrinsic signals associated with cell death [19], cell differentiation [20,21], the activation status of immune cells [22,23] and disease states in animal and human tissues [24,25,26] without labeling. The principle of Raman spectroscopy involves the inelastic scattering of light by molecules.

The majority of light is scattered at the same frequency as incident light (Rayleigh scattering). Raman scattering, which is markedly weaker than Rayleigh scattering, gains or loses energy equivalent to the allowed molecular vibrations. A Raman spectrum is obtained by measuring the number of scattered photons (light intensity) versus differences in the frequency or wavenumber between scattered light and incident light (Raman shifts). Raman spectral data offer the advantage of shape bands, which may be assigned to molecular species or functional moieties; therefore, numerous studies, including our own, have been published on Raman spectroscopy for biological applications. Investigations on reliable biomarkers to discriminate between differences in various cellular states (e.g., naïve and activated T cells and B cells) have been achieved with Raman spectroscopy combined with multivariate analyses, such as a principal component analysis (PCA) [20,21,22,23,24,25,26]. PCA is often employed for the dimensional reduction of Raman spectral data, which has thousands of data points, many of which include weak signals of interest or highly correlated information on biological states. For example, the application of the partial least square, PCA and a hierarchical component analysis to the Raman spectra of sera from healthy women and women with endometriosis enabled us to distinguish between both states [27]. Raman spectroscopy combined with multivariate analyses detected chemical changes in blood, the liver and brain caused by magnetic field exposure [28]. In the present study, we introduced fluctuations in and correlations between elements at the transition state in contradistinction to previous studies that did not focus on these fluctuations.

In the present study, we applied the DNB theory to Raman spectra of T cell activation as a model case. The combination of the DNB theory and Raman spectroscopy will provide additional information to that obtained from current multivariate analyses. We reanalyzed the Raman spectral data of T cell activation, which we previously investigated using PCA and a linear discriminant analysis (LDA) [22]. We selected data as a model case and attempted to detect the transition state and also identify DNB Raman shifts that exhibited abnormal fluctuations at the transition state. The initial and final states corresponded to naïve and fully activated T cells, respectively. Based on the DNB theory, we expected a transition state to exist during the T cell activation process and attempted to detect it.

## 2. Materials and Methods

### 2.1. Overview of the DNB Theory

Figure 1 shows a conceptual diagram of the DNB theory. Stable state 1 (the initial state) corresponds to the local minimum of the potential and is stable against disturbances (see the blue ball on the light gray line). However, when the potential well becomes very shallow at the transition state, the system becomes susceptible to disturbances and exhibits abnormal fluctuations (see the red ball). The state then moves along the slope of the potential shown by the black line and eventually settles into another state (see the green ball in stable state 2). The DNB theory aims at detecting early warning signals at the transition state by focusing on fluctuations together with a network composed of correlated elements.

The basic flow of the DNB analysis is listed below. Each detail is discussed in later paragraphs.

Preprocessing;F-test for the evaluation of fluctuations;Correlations to evaluate the relationships between fluctuating variables;Clustering to define DNB candidates;DNB scores to identify the transition state and DNB elements.

In addition to these series of procedures, specific processing is expected to be required for the biological data of interest.

Data for each time point are described as the matrix $X=x_{ik}$(i=1,…,N,k=1,…,K), where *i* and *k* are the indices of the variables and a sample of data, respectively. *N* is the total number of variables and *K* is the number of samples at the time point. The mean $m_{i}\left(X\right)$ and standard deviation $s_{i}\left(X\right)$ are defined as
(1)$$\begin{array}{ccc} m_{i}\left(X\right) & = & \frac{\sum_{k=1}^{K}x_{ik}}{K}, \end{array}$$
(2)$$\begin{array}{ccc} s_{i}\left(X\right) & = & \sqrt{\frac{\sum_{k=1}^{K}{\left(x_{ik}-m_{i}\left(X\right)\right)}^{2}}{K-1}}. \end{array}$$

The one-tailed F-test is performed to evaluate fluctuations in intensity (see Equation (Equation 2)) at each variable. The F-test is a statistical test that uses an F-distribution and its null hypothesis is that the variances of two groups being compared are the same. The data points of the two groups are assumed to independently follow normal distributions. At each time point, we select variables with significantly larger variances than the control (0 h). Multiple testing corrections using the Benjamini–Hochberg method are performed to suppress the false discovery rate. The *p* values obtained from the F-test are sorted in an ascending manner and converted to *q* values with the formula $q_{i}=\left(p_{i}N\right)/i$, where *i* is the element index and *N* is the number of elements. Variables satisfying q≤0.05 are extracted as fluctuating elements.

We then focus on the correlations among fluctuating variables that are not equally distributed. The correlation coefficient is defined as follows:(3)$$\begin{array}{ccc} r_{ij}\left(X\right) & = & \frac{\sum_{k=1}^{K}\left(x_{ik}-m_{i}\left(X\right)\right)(x_{jk}-m_{j}\left(X\right))}{\left(K-1\right)\;s_{i}\left(X\right)s_{j}\left(X\right)}, \end{array}$$
where *j* is the index of the variable different from the index *i* (j=1,…,N). A hierarchical clustering method is used to detect clusters with strong correlations among fluctuating variables. In hierarchical clustering, variables are iteratively merged according to a given metric for evaluating dissimilarity based on $r_{ij}$ in Equation (Equation 3) and a linkage method, giving a tree-like diagram called a dendrogram. The dissimilarity *d* between variables is evaluated based on $d=1-|r_{ij}|$, where $r_{ij}$ is the correlation coefficient between the *i*-th and *j*-th variables. When the correlation is strong, either positive or negative, the dissimilarity is close to zero. The average linkage method is used to evaluate the dissimilarity between tentative clusters and, accordingly, a dendrogram is plotted. We cut the dendrogram at the appropriate dissimilarity cut-off to define clusters.

To evaluate the validity of each DNB candidate group as an indicator of the transition state, the DNB score, which is the product of the average standard deviation $I_{s}$ and average correlation strength $I_{r}$ [29], is calculated as follows:(4)$$I_{\mathrm{DNB}}=I_{s}\cdot I_{r}.$$

A set of element indices of a DNB candidate group is denoted as $S^{*}$ and $|S^{*}|$ denotes its size. The average standard deviation $I_{s}$ and the average correlation strength $I_{r}$ are defined for each time point’s data matrix *X* as follows: (5)$$\begin{array}{ccc} I_{s} & = & \frac{1}{\left|S^{*}\right|}\sum_{i\in S^{*}}s_{i}\left(X\right), \end{array}$$(6)$$\begin{array}{ccc} I_{r} & = & \frac{2}{\left|S^{*}\right|\left(\left|S^{*}\right|-1\right)}\sum_{i,j\in S^{*},\;i<j}^{}\left|r_{ij}\left(X\right)\right|, \end{array}$$
where *i* and *j* are different indices. Since the F-test and hierarchical clustering are performed at each time point, different DNB candidate groups are obtained for different time points. We plotted the time evolution of $I_{s}$, $I_{r}$ and $I_{\mathrm{DNB}}$, defined by Equations (Equation 4)–(Equation 6), for each DNB candidate group. If the DNB score showed a peak at the same time point when the DNB candidate group was extracted, the time point was regarded as the transition state and the corresponding elements were taken as DNB elements. We also investigate whether $I_{s}$ and $I_{r}$ both took high values at the time point.

Caution is needed when using the original DNB score [1] because Raman spectra contain noise that cannot be ignored. Real-world data may contain noise or the system may be divided into disconnected subgroups. Therefore, the redefined DNB score [29], described by Equation (Equation 4), was used to avoid erroneous behavior.

### 2.2. Raman Spectra of the T Cell Activation Process

T cells are a class of immune cells that are involved in acquired immunity. They are initially in a naïve state and are activated by antigen-presenting cells or stimulating factors via the αβ-T cell receptor (TCR complex) and a co-stimulation, which is important for antigen-induced activation. After being activated, T cells rapidly replicate and differentiate to become effector T cells and this process generally takes 1–2 days. We extracted naïve CD4${}^{+}$ T cells from splenic suspensions [22] obtained from DO11.10 TCR transgenic mice on the BALB/c background. T cells were cultured on a plate pre-coated with antibodies against CD3 and CD28 (1 μg/mL each, from BD Biosciences, Franklin Lakes, NJ, USA). All Raman spectra used in the T cell analysis were those from a previous study [22].

Raman spectra were obtained with a home-built line scanning microscope based on a Nikon Ti microscope (Nikon, Tokyo, Japan) equipped with a spectrometer (MK-300; Bunkoh Keiki, Tokyo, Japan) as previously reported. T cells in Tyrode’s solution were placed on silica coverslips (SPI supplies, West Chester, PA, USA) and observed through a 40 × water immersion objective lens with a 1.27 numerical aperture (Nikon CFP Plan Apo IR; Nikon). The sample was illuminated with a 532 nm laser at 2.4 mW/$\mu m^{2}$. Cells that died after observations were identified on a bright field image and removed from the analysis.

Data used in the present study were the same as those previously reported [22] with slight modifications in the pre-processing of the spectrum. The time points examined were 0 (the naïve state of T cells), 2, 6, 12, 24 and 48 h (fully activated T cells). Regarding pre-processing, the cosmic ray was removed from spectra and the cell region was examined using 1008 ${\mathrm{cm}}^{-1}$ intensity by Otsu’s binarization method [30]. Spectra from the cell region were then averaged to obtain the spectrum of the cell, while spectra from the non-cell region were subtracted as the background signal. The cell area was also calculated by counting the number of pixels in the cell. The Raman shift axis was calibrated using peaks at 753 and 1587 ${\mathrm{cm}}^{-1}$ with respect to measurement conditions, such as temperature and linearly interpolated between the fingerprint region 500–1799 ${\mathrm{cm}}^{-1}$ at 1 ${\mathrm{cm}}^{-1}$ intervals. Residual auto-fluorescence was removed using a rolling-ball algorithm [31]. In this algorithm, a circular or elliptic ball is placed below a Raman spectrum’s curve with the horizontal axis of Raman shifts and the vertical axis of Raman intensities. By moving the ball to keep touching the spectrum’s curve, a smooth baseline curve is calculated and removed. This algorithm has the following two parameters: the ball’s major diameter along the horizontal axis *a* and ellipticity ε=b/a, where *b* is the minor diameter along the vertical axis. We considered six values for the major diameter from 500 to 1000 in 100 increments and three values (ε = 0.05, 0.5 and 1.0) for ellipticity. By combining these parameter values, 18 cases were considered. The Raman signal was normalized by the sum intensity between 500 and 1799 ${\mathrm{cm}}^{-1}$. The intensity of each Raman shift was divided by the average intensity from 548 to 1799 ${\mathrm{cm}}^{-1}$. Raman shifts below 548 ${\mathrm{cm}}^{-1}$, which contained glass signals, were only excluded in calculations of the mean value.

### 2.3. PCA-LDA

Traditional PCA and LDA (PCA-LDA) were used to distinguish between the naïve and activation states of T cells. In the present study, 1300 types of Raman shifts were reduced to eight dimensions through PCA. A model was then constructed to classify the naïve state at 0 h and fully activated states at 48 h using LDA. The discriminant score was also calculated from the degree of classification and used as a biomarker for T cell activation [22].

### 2.4. A DNB Analysis Suitable for Raman Spectra

We assumed that the naïve state corresponded to stable state 1 and the activation state to stable state 2 in Figure 1. The transition state may exist from the naïve state to the activation state according to the DNB theory. To apply the DNB theory to Raman spectra, we assigned Raman shifts to variables $X=x_{ik}$ and Raman spectral intensities to the realization of variables in the complex network underlying the DNB theory. The DNB analysis was performed according to the basic flow described in Section 2.1.

In the procedure of the F-test, the control group was data in the initial states: 0 h for the activation processes. All other data at 2, 6, 12, 24 and 48 h were set as the experimental groups. The value q≤0.05 converted by the Benjamini–Hochberg method was used to extract fluctuating Raman shifts. In hierarchical clustering, we cut the dendrogram at the dissimilarity cut-off of 0.3 to define clusters. We then selected the largest cluster and other relatively large clusters that were at least half the size of the largest one. Conversely, clusters with too few elements were removed as candidates. When the Raman shifts obtained within a cluster were continuous, we considered them to be a spurious correlation and removed them from the DNB candidate group. This was because, for example, if Raman shifts are continuous as $\left\{x_{i-2},\;x_{i-1},\;x_{i},\;x_{i+1},\;x_{i+2}\right\}$, we cannot exclude the possibility that Raman intensity at the center $x_{i}$ is convolved into the surrounding area $x_{i\pm 1},\;x_{i\pm 2}$ due to the characteristics of Raman spectroscopy.

In the preprocessing of Raman spectral data, peak filtering was additionally performed only when representative peaks were being analyzed. In peak filtering, we selected peak positions at which the center of three consecutive points in data at 6 h was the largest. The number of peak positions was 81 points and the peaks obtained were as follows: 509, 520, 523, 529, 536, 549, 561, 566, 572, 579, 606, 608, 632, 648, 650, 661, 684, 687, 735, 751, 755, 771, 790, 813, 832, 849, 854, 857, 869, 896, 902, 937, 955, 968, 984, 986, 1003, 1006, 1030, 1054, 1064, 1093, 1095, 1126, 1129, 1150, 1158, 1171 1173, 1254, 1307, 1310, 1322, 1334, 1337, 1368, 1370, 1373, 1446, 1474, 1482, 1485, 1523, 1576, 1609, 1616, 1619, 1654, 1656, 1728, 1730, 1737, 1739, 1747, 1751 1755, 1761 1766, 1774, 1780 and 1793 ${\mathrm{cm}}^{-1}$.

## 3. Results and Discussion

### 3.1. Conventional Analysis

Prior to performing the DNB analysis, we conducted a conventional analysis to obtain an overview of data. We attempted here to visualize the temporal features of typical Raman shifts and to introduce PCA-LDA.

Figure 2 shows changes in Raman spectra for the T cell activation process. Raman spectral intensities averaged across cells at each time point produced by the raw and preprocessed datasets are shown in Figure 2a,b, respectively. A comparison of both panels showed that preprocessing including a rolling-ball algorithm worked well. Several characteristic peaks were observed in the Raman spectrum (see five inverted triangles in Figure 2b). Typical peaks were attributed to specific molecular vibrations. For example, the peak located at Raman shift 1004 ${\mathrm{cm}}^{-1}$ was attributed to a phenylalanine ring breath. The intensities of typical peaks varied with time. In Figure 2c–g, the Raman intensities of (c, e) cytochrome C, (d) the phenylalanine ring breath, (f) DNAs and (g) lipids/proteins did not monotonically increase or decrease over time, suggesting that none had a one-to-one correspondence with the state of T cell activation. On the other hand, the average cell area as shown in Figure 2h monotonically increased over time, except for at 6 h and markedly increased between 12 and 24 h, which is consistent with our previous findings [22]. Therefore, the cell area is a good indicator of T cell activation and a major phenotypic change occurred between 12 and 24 h. In addition, phenylalanine and the cell area took a minimum value at 6 h, while DNAs and lipids/proteins peaked at the same time point. These results suggest that T cell activation is not a monotonic process, it involves transitions between distinct stages.

Figure 3 shows the classification of naïve and activation states by PCA-LDA. The classifier was a model that distinguishes between naïve (0 h) and activation states (48 h). A significant difference was not initially observed between the naïve state (0 h) and the state after a short time (2 h), as shown in Figure 3a. As time passed, naïve and other data become separate around an LDA score of zero. It was difficult to identify transition states through the changes observed in Figure 3a–e. Therefore, a different method was required to detect the transition state during the T cell activation process. As shown in Figure 3f, the discriminant score gradually increased over time, which is consistent with the cell area as a biomarker (see Figure 2h).

### 3.2. The DNB Analysis Using Full-Range Raman Shifts

We performed two types of DNB analyses: one using full-range Raman shifts and another using peak-filtering Raman shifts. The results of the first analysis are shown in this section. The results of a DNB analysis generally depend on the details of data preprocessing. In the present study, we investigated the two parameters of the rolling-ball algorithm and 18 parameter sets were produced. DNB candidate groups were obtained at 2, 6 and 12 h. The cluster size of the DNB candidate group obtained at 2 h was too small to discuss specified attributions to materials and compositions. The DNB candidate group obtained at 12 h was regarded as invalid because the DNB score had not peaked at the time point. At 6 h, a wide variety of DNB candidate groups were obtained in a manner that depended on rolling-ball parameters and they were verified by the DNB score. We herein focused on DNB Raman shifts obtained at 6 h with a major diameter of 1000 and ellipticity of 0.05 as an example.

Figure 4 shows the results of the DNB analysis using full-range Raman shifts from 500 to 1799 ${\mathrm{cm}}^{-1}$. We extracted 163 Raman shifts showing significantly large fluctuations from 1300 types of Raman shifts at 6 h using the F-test and Benjamini–Hochberg method. The hierarchical clustering method was then applied, producing the dendrogram shown in Figure 4a. The largest cluster (with the largest number of elements shown in the light blue part) was extracted as a DNB candidate group. The other clusters were excluded from the analysis because they were smaller than half of the number of elements in the largest cluster. We obtained a DNB candidate group consisting of 81 Raman shifts: 693, 701 707–708, 718–721, 724–733, 735–740, 742–759, 1226–1228, 1234–1237, 1239–1247, 1249–1261, 1267–1273 and 1276–1278 ${\mathrm{cm}}^{-1}$ (consecutive Raman shifts are hyphenated for simplicity). Note that the 81 DNB candidates coincidentally matched the number of representative Raman shifts in the preprocessing of the peak-filtering method, but were completely different.

As shown in Figure 4b, the average discriminant score gradually increased as a biomarker of the T cell activation process. Naïve T cells varied and then became fully activated until 48 h. On the other hand, the DNB score peaked at 6 h and then decreased, reaching a minimum at 48 h. The average standard deviation had large values at 6 and 12 h and the average correlation strength peaked at 6 h (see Figure 4c,d). Therefore, we concluded that 6 h was the transition state for the T cell activation process and the 81 related Raman shifts were identified as DNB elements. Figure 4e shows the topological structures of the weighted correlation network of DNB Raman shifts. Each Raman shift is represented as a node and the correlation coefficient between Raman shifts as an edge. Positive and negative correlation intensities are shown in red and blue, respectively. However, since the information in this figure was difficult to interpret, we attempted to visualize it using a color map. Figure 4f shows the time evolution of correlation coefficients between DNB Raman shifts. We found two groups consisting of Raman shifts 693–759 and 1226–1278 ${\mathrm{cm}}^{-1}$. The correlation strength between DNB Raman shifts, whether positive or negative, was the highest at 6 h and the lowest at 48 h, which is consistent with the line plot of the average correlation strength (see Figure 4d). Furthermore, Raman shifts within each group positively correlated and these correlations had similar strengths. The two groups also showed negative correlations with each other. These results indicate an underlying interaction network associated with DNB Raman shifts, the strength of which varied in time and peaked at the transition state.

The time points available in the present study were 0, 2, 6, 12, 24 and 48 h, as described in a previous study. We identified the transition state within several limited time points. Nevertheless, the selection of an appropriate time interval will be pivotal in future experiments. Our proposed strategy for the time interval is 2 h because each measurement by Raman spectroscopy takes several tens of minutes and the preparation and clean-up of equipment are required.

### 3.3. The DNB Analysis Using Peak-Filtering Raman Shifts

We performed a second DNB analysis using peak-filtering Raman shifts instead of full-range Raman shifts. It is important to note that the DNB analysis using full-range Raman shifts violated the assumption that variables are independent, which may lead to incorrect results because the signals were convoluted to the surrounding Raman shifts due to the instrument function of the spectrometer. Similarly, data smoothing in preprocessing is another point of caution. Therefore, a second analysis was performed to address this issue.

Using the procedure described in the preprocessing of Raman spectral data and peak filtering sections, we extracted 81 representative peaks from 1300 types of Raman shifts. By applying the F-test and Benjamini–Hochberg method, 10 fluctuating Raman shifts were obtained at 6 h.

Figure 5 shows the results of the DNB analysis using peak-filtering Raman shifts. This figure is shown in a similar manner to Figure 4. In Figure 5a, the dendrogram was so simple that the largest cluster consisted of 735, 751, 755 and 1254 ${\mathrm{cm}}^{-1}$, which were taken as DNB candidates (see the orange part). In Figure 5b–d, the DNB score, average standard deviation and average correlation strength peaked at 6 h. Therefore, we concluded that the transition state of the T cell activation process occurred at 6 h and Raman shifts of 735, 751, 755 and 1254 ${\mathrm{cm}}^{-1}$ were identified as DNB elements. These four Raman shifts were also included in the 81 DNB Raman shifts obtained when using full-range Raman shifts. Figure 5e shows the topological structures of the weighted correlation network of four DNB Raman shifts. Three out of the four nodes positively correlated with each other. Similarly, as shown in Figure 5f, we found that two groups negatively correlated. In a comparison with Figure 5d, these correlations were the strongest at 6 h, while no correlation was observed at 48 h.

Possible assignments for Raman shifts identified as DNBs in biological tissues were given as follows: A Raman shift of 735 ${\mathrm{cm}}^{-1}$ was assigned to a protein-derived C-S bond. Raman shifts of 751 and 755 ${\mathrm{cm}}^{-1}$ were relatively close to and included in the symmetric breathing of tryptophan in the 752–755 ${\mathrm{cm}}^{-1}$ range, respectively. The remaining Raman shift, 1254 ${\mathrm{cm}}^{-1}$, was assigned to C-N plane stretching. Other possibilities include lipids (1255 ${\mathrm{cm}}^{-1}$) or a slightly shifted Amide III (1257–1260 ${\mathrm{cm}}^{-1}$), which also exhibits C-N stretching or N-H bending vibrations [32].

We investigated the two types of DNB analyses using full-range and peak-filtering Raman shifts and reached a consistent conclusion that the transition state was at 6 h. In addition, Figure 4b–d and Figure 5b–d showed a very similar time evolution in all quantities, except for the average standard deviation at 48 h. These results suggest that although nearby Raman shifts correlated, full-range Raman shifts may be used to detect the transition state of the T cell activation process. A DNB analysis using full-range Raman shifts is simple and suitable, while a targeted analysis using peak-filtering Raman shifts is also beneficial. The advantage of using peak-filtering Raman shifts is the need for fewer computer resources because the number of data points, such as Raman shifts, is markedly lower than that using full-range Raman shifts. However, the use of peak-filtering Raman shifts requires an extra step to prepare representative peaks from Raman spectra in advance. Therefore, a DNB analysis using full-range Raman shifts without the need for peak filtering was better for detecting the transition state.

## 4. Conclusions

In the present study, we applied the DNB theory to the Raman spectra of the T cell activation process and successfully detected the early warning signals of a transition state at 6 h with or without peak filtering. Raman intensity is convoluted into the surrounding shifts due to the instrument function of the spectrometer, making it difficult to apply the DNB theory using independent variables. However, an important point for detecting the early warning signals of transition states is the simplicity of using full-range Raman shift data. Furthermore, the transition state of the T cell activation process is 6 h and the average discriminant score in conjunction with the average cell area markedly increased between 12 and 24 h. This indicates that a hidden or non-phenotypic transition occurred immediately after 6 h and triggered later cell expansion. The view of 6 h as a turning point is also supported by the time evolution of the average intensities of phenylalanine, DNAs and lipids/proteins. Future studies are warranted to clarify why DNB Raman shifts change depending on the parameters of the rolling-ball algorithm and also to improve our data analysis pipeline for more robust detection.

In conclusion, the present results suggest novel applications of the DNB theory to Raman spectra, obtained with Raman spectroscopy, based on the label-free and non-invasive detection of early warning signals at the transition state in living cells or tissues. In fundamental research on cellular mechanisms, a number of applications will be possible for detecting the transition state in the process of not only cell activation, but also differentiation and apoptosis. Furthermore, based on the identification of the transition state on blood cells (T cells), we will detect the transition state (pre-disease state) between healthy and disease states using Raman spectroscopy and related DNB Raman shifts in future clinical studies. Our approach will also be useful for obtaining a more detailed understanding of the biological mechanisms of the transition state (pre-disease state) when molecular compositions, molecular structures and chemical bonding attributed to the identified DNB Raman shift groups are clarified.

## Figures and Tables

**Figure 1 biomolecules-12-01730-f001:**
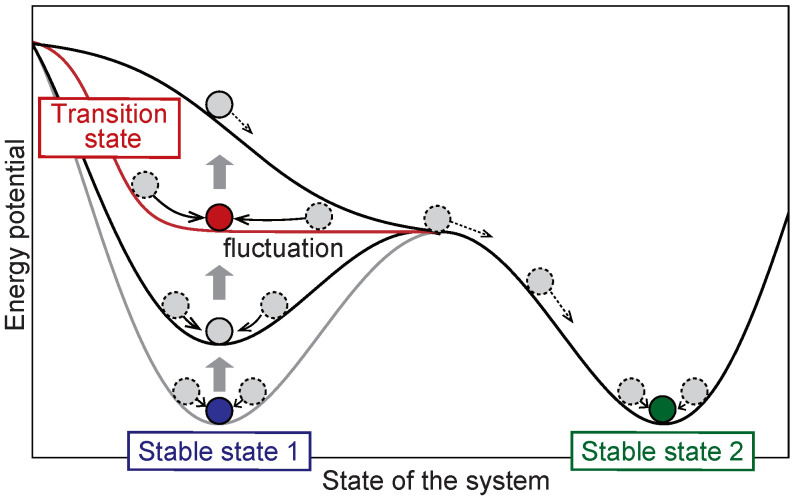
A conceptual diagram of the DNB theory.

**Figure 2 biomolecules-12-01730-f002:**
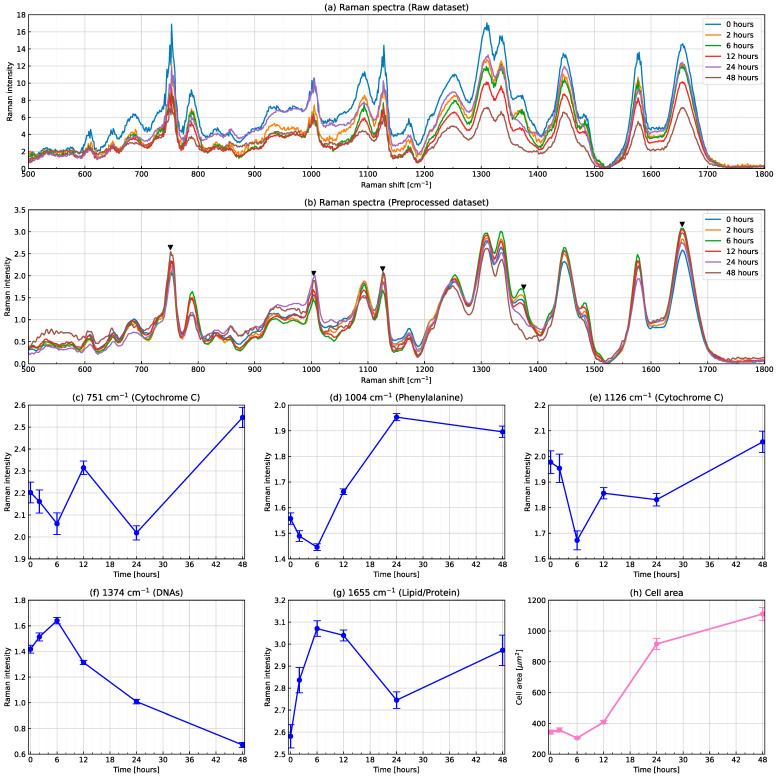
Changes in Raman spectra for the T cell activation process. Raman spectral intensities averaged across cells at each time point produced by (**a**) raw and (**b**) preprocessed datasets. Time evolution of Raman intensities at (**c**,**e**) cytochrome C, (**d**) the phenylalanine ring breath, (**f**) DNAs and (**g**) lipids/proteins and (**h**) the average cell area. Error bars show the standard deviation in panels (**c**–**h**).

**Figure 3 biomolecules-12-01730-f003:**
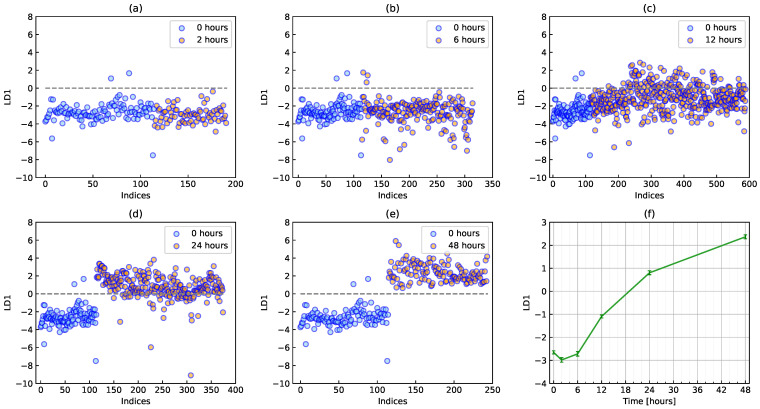
Classification of naïve and activation states by PCA-LDA. Scatter plots of discriminant scores at (**a**) 2, (**b**) 6, (**c**) 12, (**d**) 24 and (**e**) 48 h together with the naïve state at 0 h. Red circles show cells at each time point together with blue circles at 0 h. (**f**) Time evolution of the average discriminant score during T cell activation. Error bars show the standard deviation.

**Figure 4 biomolecules-12-01730-f004:**
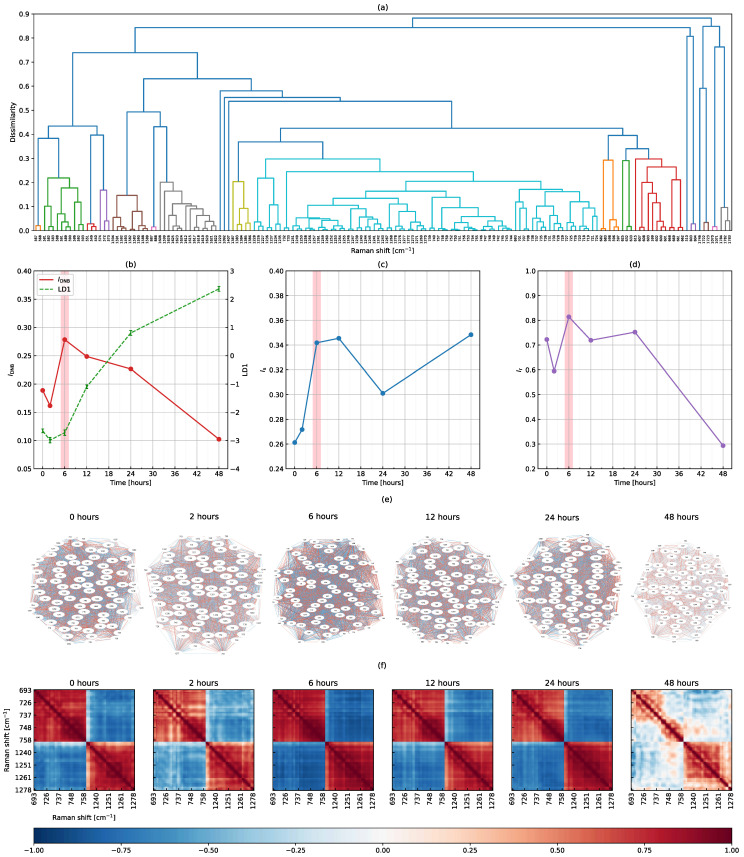
Results of the DNB analysis using full-range Raman shifts. (**a**) A dendrogram produced by the hierarchical clustering of 163 Raman shifts fluctuating at 6 h, the time evolution of (**b**) the DNB score overplotted with the average discriminant score, (**c**) the average standard deviation and (**d**) the average correlation strength among DNB candidates, (**e**) topological structures of the weighted correlation network, and (**f**) correlation coefficients between 81 DNB Raman shifts. In (**b**), the average discriminant score is shown again as the dashed line from Figure 3f.

**Figure 5 biomolecules-12-01730-f005:**
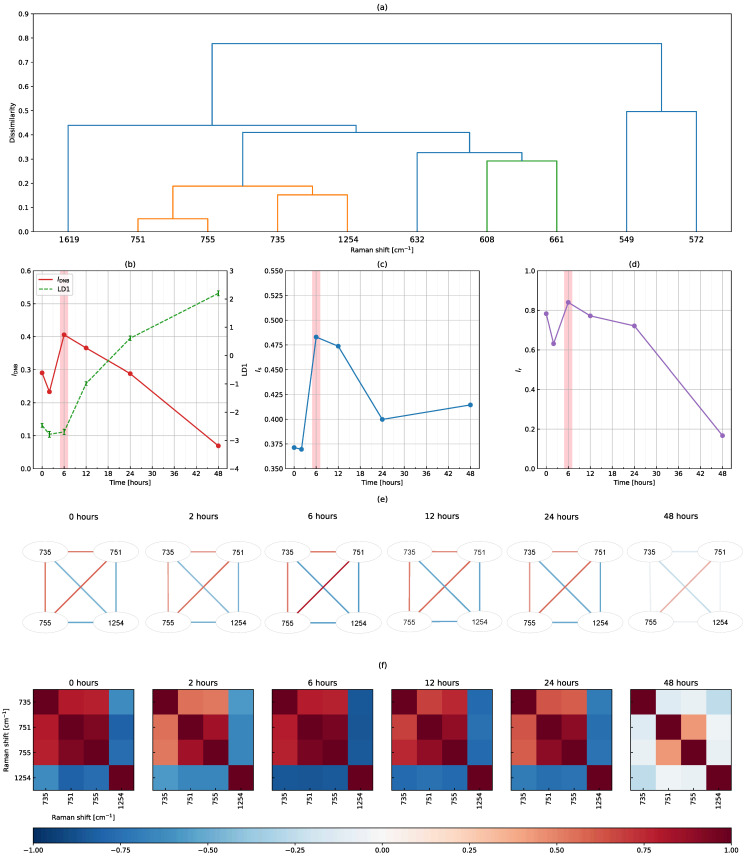
Results of the DNB analysis using peak-filtering Raman shifts. (**a**) A dendrogram produced by the hierarchical clustering of 10 Raman shifts fluctuating at 6 h, the time evolution of (**b**) the DNB score overplotted with the average discriminant score, (**c**) the average standard deviation and (**d**) the average correlation strength among DNB candidates, (**e**) topological structures of the weighted correlation network, and (**f**) correlation coefficients between the four DNB Raman shifts.

## Data Availability

The data presented in this study are available on request from the corresponding authors.

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
