# Peer review of "Application of the Dynamical Network Biomarker Theory to Raman Spectra"

_biomolecules, 2022, doi:10.3390/biom12121730_

Round 1
Reviewer 1 Report
In manuscript biomolecules-1983925 entitled “Application of the dynamical network biomarker theory to Raman spectra”, authors proposed to use DNB model and method to study the tipping point event during temporal molecule data from Raman spectroscopy.
This is a relevant work of DNB application using non-destructive testing biological information.
I have a few suggestions:
(1) The recent developments of DNB were missed in introduction, e.g. the single-sample DNB or DNB landscape (PMID: 34691933, PMID: 34609489, etc).
(2) There are different measurements of DNB score, it is suggested to discuss and evaluate the score authors used as ref.24 and that score in ref.1.
(3) As a new data from Raman spectroscopy, it is better to demonstrate the original data of Raman spectroscopy and preprocessed data, which help data understanding for readers in different fields.
(4) This work used a temporal data at different time points, it is better to introduce and discuss the design of time interval of this data, e.g. how to guarantee the observation or detection of tipping point at the given time points.
(5) For the identified key point, e.g. 6h, it is necessary to analyze more details of the molecular species or functional moieties underlying Raman spectroscopy, so as to supply more biological evidences for this early-warning point.
(6) For Fig4.E and Fig5.E, it is suggested to illustrate the weighted topological structure of correlation network, in addition to heatmap, which should clearly display the characteristics of DNB identified, as network biomarker.
Author Response
We thank you for your comments, which have improved the quality of the revised manuscript. Please find our revised manuscript and response letter.

Reviewer 2 Report
1. This paper is very well written and full of work. I suggest the authors to make minor revisions. In addition, it is suggested that the introduction and methods can be enriched, and the following papers are recommended:
1. Detection of the chemical changes in blood, liver, and brain caused by electromagnetic field exposure using Raman spectroscopy, biochemical assays combined with multivariate analyses
2. Correlation between endometriomas volume and Raman spectra. Attempting to use Raman spectroscopy in the diagnosis of endometrioma
Author Response
We appreciate your positive comments on our manuscript. Please find our revised manuscript and response letter.

Round 2
Reviewer 1 Report
Authors have responded to all my concerned questions, and made a good revision.